# Unfolding recurrence by Green's functions for optimized reservoir computing

**Sandra Nestler**[1,2,5], **Christian Keup**[1,5], **David Dahmen**[1],
**Matthieu Gilson**[1,4], **Holger Rauhut**[2], **Moritz Helias**[1,3]
[1]Institute of Neuroscience and Medicine (INM-6), Jülich Research Centre, Jülich, Germany
[2]Mathematics of Information Processing, RWTH Aachen University, Aachen, Germany
[3]Department of Physics, RWTH Aachen University, Aachen, Germany
[4]Universitat Pompeu Fabra, Barcelona, Spain
[5]RWTH Aachen University, Aachen, Germany
{s.nestler,c.keup,d.dahmen,m.gilson,m.helias}@fz-juelich.de,
rauhut@mathc.rwth-aachen.de

## Abstract

Cortical networks are strongly recurrent, and neurons have intrinsic temporal dynamics. This sets them apart from deep feed-forward networks. Despite the tremendous progress in the application of feed-forward networks and their theoretical understanding, it remains unclear how the interplay of recurrence and non-linearities in recurrent cortical networks contributes to their function. The purpose of this work is to present a solvable recurrent network model that links to feed forward networks. By perturbative methods we transform the time-continuous, recurrent dynamics into an effective feed-forward structure of linear and non-linear temporal kernels. The resulting analytical expressions allow us to build optimal time-series classifiers from random reservoir networks. Firstly, this allows us to optimize not only the readout vectors, but also the input projection, demonstrating a strong potential performance gain. Secondly, the analysis exposes how the second order stimulus statistics is a crucial element that interacts with the non-linearity of the dynamics and boosts performance.

## 1 Introduction

Trained neural networks today form an integral component of data science. Widely used approaches comprise deep neural networks [LeCun, 2015] that typically employ time-independent mappings by hierarchical structures with mostly feed-forward connections. In contrast, recurrent neural networks, which follow more closely their biological counterparts in the brain, have units with intrinsic temporal dynamics that allow natural processing of time-dependent stimuli. The interplay of recurrence and non-linearity in such networks renders their analysis challenging. There is large interest in understanding the basis for their computational abilities. Reservoir computing, as originally introduced via Echo State Networks [Jaeger, 2001] and Liquid State Machines [Maass et al., 2002], is one approach that takes recurrence of connections and temporal dynamics into account. Signals are here mapped into a high dimensional space spanned by a large number of typically randomly connected neurons, on which a linear readout is trained. The network thereby acts like a kernel in a support vector machine [Vapnik, 1998, Cortes and Vapnik, 1995]. The training can be combined with a feedback of the readout signal to effectively modify also the recurrent connections [Sussillo and Abbott, 2009, DePasquale et al., 2018]. The gradient of an arbitrary loss function for these models can be computed memory efficiently via ordinary differential equations [Chen et al., 2018]. Although recurrent models lately have become more and more complex [Hochreiter and Schmidhuber, 1997, Cho et al., 2014, Collins et al., 2016], they remain highly similar to simple reservoirs in terms of the

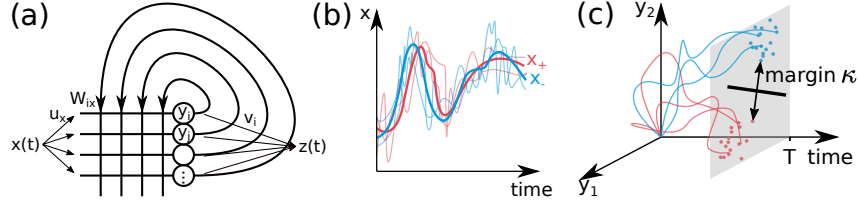

Figure 1: *Binary classification with recurrent dynamics.* (a) A neural network with random connectivity $W$ is stimulated with an input $x(t)$ via an input vector $u$ (left). A linear readout with weights $v$ transforms the high dimensional state into a scalar quantity $z(t)$. (b) Time course of sample stimuli (colored thin curves) from two different classes (red, blue; thick curves: class average). In this example, classes differ mainly in fluctuations. (c) Responses of the network follow high-dimensional trajectories (colored curves, only two dimensions $y_1, y_2$ shown for conceptual clarity). At readout time $T$, the samples form clouds of states, indicated by points in the readout time plane. Classification places a decision plane between the classes. The margin $\kappa$ is the smallest distance between the states and the plane.

learned neural representations [Maheswaranathan et al., 2019]. Furthermore, it has been extensively studied how the performance of the reservoir depends on the properties of the recurrent connectivity; the edge of chaos has been found as a global indicator of good computational properties [Bertschinger et al., 2005, Toyoizumi and Abbott, 2011]. However, the interplay of recurrence and non-linearities may, depending on the statistical features of the input data, offer optimal settings that are not described by such global parameters alone.

We here set out to systematically analyze the kernel properties of recurrent time-continuous networks in a binary time series classification task. We show how the high-dimensional and non-linear transformation implemented by the network can be used to selectively extract differences in the statistics between a pair of input classes. To this end, we analyze the mapping between the input data distribution and the shape and linear separation of the resulting network states, which uniquely determine the optimal readout projection. In state-of-the-art reservoir computing, the projection of the stimuli into the network is mostly carried out with random weights. To the contrary, we here show that the classification performance crucially depends on the input projection; random projections consistently lead to significantly sub-optimal performance, whereas an optimal input projection exploits the mode landscape of the reservoir to obtain an advantageous configuration of the resulting distribution of network states. We derive a method to jointly optimize both projections in a system of linear units and generalize these results to non-linear networks. To this end, we employ a perturbative approach that transforms the non-linear recurrent network into an effective feed-forward structure. The analytical expressions expose how the network dynamics separates a priori linearly non-separable time-series. We find that even weak non-linearities can significantly boost the separability of network states if the linear separability of the stimuli is low.

## 2 Setup

We consider a reservoir model shown in figure 1(a): A time-dependent input function $x(t)$ is projected into the $N$-dimensional neuron space with input projection $u \in \mathbb{R}^N$. This signal reverberates in the network through continuous interactions via recurrent connections $W \in \mathbb{R}^{N \times N}$ as well as sustained external stimulation, leading to a neural trajectory $y(t) \in \mathbb{R}^N$ that is described by the first-order differential equation [Sompolinsky et al., 1988]

$$(\tau \partial_t + 1)\, y_i(t) = \sum_j W_{ij} \phi(y_j(t)) + u_i x(t), \tag{1}$$

where $\phi$ is the (non-linear) gain function of the neurons. The network activity is read out linearly by the one-dimensional projection $z(t) = v^{\mathrm{T}} y(t)$, obtained with readout vector $v \in \mathbb{R}^N$. We here consider fixed realizations of i.i.d. weights $W_{ij} \sim \mathcal{N}(0, \frac{g^2}{N})$ denoting the connections from neurons $j$ to neurons $i$ and aim towards a joint optimization of input and readout projections $u$ and $v$, respectively. In general, the existence of optimal projection vectors allows one to first define and second study the performance of the recurrent reservoir itself. Thus, common methods for optimizing

recurrent connectivity can be combined with our algorithm to study and improve the kernel properties of a reservoir network, eliminating variability of performance caused by sub-optimal input and readout projections.

Consider inputs from two classes $+$, $-$ defined by their underlying statistics, for example their mean trajectories and fluctuations, as shown in figure 1(b). The network transforms the differences across classes into distinct sets of network states $y_\pm(t)$, which form extended clouds in state space due to intra-class variability (figure 1(c)). For classification, the network space is divided by a hyperplane into one region for each class. Position and orientation of this plane are modified by the training algorithm of the readout projection $v$, the hyperplane's normal vector. The margin, the distance between the plane and the sample state closest to it, is hereby a typical optimization objective [Vapnik, 1998, Cortes and Vapnik, 1995].

## 3 Linear Networks

To introduce the concepts, we first investigate the benefits of optimized input projections for linear reservoirs, where $\phi(y) = y$ in equation (1). The linear equation of motion has the Green's function [Risken, 1996]

$$G^{(1)}(t, t') = H(t - t') \frac{1}{\tau} \exp\left[-(\mathbb{I} - W)\frac{t - t'}{\tau}\right], \tag{2}$$

where $H$ is the Heaviside function. The state of neuron $i$ at time point $t$ is then given by

$$y_i(t) = \sum_p \int_{-\infty}^{\infty} \mathrm{d}t'\, G_{ip}^{(1)}(t, t')\, u_p\, x(t'). \tag{3}$$

The margin between classes of stimuli with class labels $\zeta_\nu \in \pm 1$

$$\kappa(u, v) = \min_\nu (\zeta_\nu v^{\mathrm{T}} y^{u,\nu}), \tag{4}$$

where $v$ has unit length, constitutes a measure to be optimized to increase generalization performance. We here denote by $y^{u,\nu}$ the network response to stimulus $x^\nu$ projected via input vector $u$, and we assumed that the separating hyperplane passes through the origin. This choice is adequate for the stimulus set employed below. Shifting the plane off the origin can be accounted for by incorporation of a threshold. The margin $\kappa$ depends on both the input projection $u$ and the readout $v$. For a given set of training data, its maximum is uniquely defined by the support vector machine algorithm [Vapnik, 1998, Cortes and Vapnik, 1995]. For the joint optimization of input and readout projections we pursue here, we use this objective as the basis to derive analytically tractable approximations.

For generality of the optimal projection vectors and analytical insight, it is advisable to replace the minimum function in equation (4) by a differentiable approximation, leading us to a soft margin which takes into account not only the outliers, but all points weighted by their distance to the classification plane. This has the advantage to tolerate some outliers if this improves the distance for the majority of samples that are closer to the plane. Here we use a soft margin of the form [Lange et al., 2014]

$$\kappa_\eta(u, v) = -\frac{1}{\eta} \ln\left[\sum_\nu \exp(-\eta \zeta_\nu v^{\mathrm{T}} y^{u,\nu})\right]. \tag{5}$$

The control parameter $\eta$ regulates the importance of distances of states close to and far from the separating hyperplane. For $\eta \to \infty$, we recover the margin $\kappa = \lim_{\eta \to \infty} \kappa_\eta$. For finite $\eta$, the soft margin becomes less sensitive to the exact realizations of the network states than the margin $\kappa$ (equation (4)). We show in the supplementary material (section A.1) by Hölder's inequality that equation (5) is in fact concave in $v$; it thus possesses a unique maximum in $v$. As we presume a large number of samples representing the distribution of stimuli, we can express the sum by an expectation value with respect to the underlying probability distribution of $\zeta_\nu y^{u,\nu}$,

$$\kappa_\eta(u, v) \to -\frac{1}{\eta} \ln\left\langle \exp(-\eta \zeta_\nu v^{\mathrm{T}} y^{u,\nu})\right\rangle,$$

where we neglected an inconsequential offset. The soft margin $\kappa_\eta$ has now the form of a scaled cumulant generating function [Gardiner, 1985, Touchette, 2009]; its Taylor expansion until the second cumulant of $\zeta_\nu y^{u,\nu}$ thus reads

$$\kappa_\eta(u, v) \approx v^{\mathrm{T}} M^u - \frac{1}{2}\eta\, v^{\mathrm{T}} \Sigma^u\, v, \tag{6}$$

where $M_i^u := \langle \zeta_\nu y_i^{u,\nu} \rangle$ is the average separation vector between the center of the clouds and the decision plane and $\Sigma_{ij}^u := \langle (\zeta_\nu y_i^{u,\nu})(\zeta_\nu y_j^{u,\nu}) \rangle - M_i^u M_j^u$ the covariance matrix. The two terms have counteracting effects on the soft margin. The decomposition of the soft margin into cumulants of labeled network states shows a suppression of cumulants of order $k$ by a factor $\frac{1}{k!}$. It is also geometrically plausible that lower order cumulants are more important than higher orders; they describe the rough shape of the state clouds. Stopping after second order amounts to a Gaussian approximation of the state clouds. Alternatively, one can regard equation (6) as classification by linear discriminant analysis if the two sample classes are of equal size [Minasny, 2009]; when further assuming Gaussianity and equal variance of the two classes, this is identical to Fisher linear discriminant analysis. From now on, the term soft margin will refer to equation (6).

Since a linear gain function $\phi(y) = y$ imposes a linear relationship between network inputs and outputs (equation (3)), each cumulant of the network state depends only on the corresponding cumulant of the stimulus. Separation between the classes is thus linearly related to the difference between mean stimuli of the two classes. In contrast, in non-linear networks higher order cumulants also contribute to the separation between the classes.

Optimization of the soft margin can be done for arbitrary input signals. However, since equation (6) only depends on the mean and covariance of network outputs, it is sufficient for linear networks to regard stimuli as coming from a Gaussian distribution. As an example, in the following the stimuli are furthermore taken as step-wise constant, accounting for a finite temporal resolution $\Delta t$ that would typically appear in a practical application. We therefore replace the dependence on the stimulus time $t'$ in the Green's function by the index $n$, where $t_n = n\,\Delta t$, defining $G_{ipn}^{(1)}(t) := \int_{t_n}^{t_{n+1}} G_{ip}^{(1)}(t,t')\,dt'$. Without loss of generality we can assume the distribution of stimuli to be of the form

$$x^\pm \propto \mathcal{N}(\pm\mu, \psi \pm \chi), \qquad (7)$$

where $\mu \in \mathbb{R}^{T/\Delta t}$ and $\psi, \chi \in \mathbb{R}^{T/\Delta t \times T/\Delta t}$. A potential offset in the mean could be absorbed by a corresponding threshold in equations (4) - (6) and different covariances $C^\pm$ are included by setting $\psi := \frac{1}{2}(C^+ + C^-)$ and $\chi := \frac{1}{2}(C^+ - C^-)$. It is straightforward to then compute the average separation at time $T$

$$M_i^u = \sum_{p,n} G_{ipn}^{(1)}(T)\,u_p \mu_n$$

and the covariance

$$\Sigma_{ij}^u = \sum_{n,m,p,q} G_{ipn}^{(1)}(T)\,G_{jqm}^{(1)}(T)\,u_p u_q\,\psi_{nm}.$$

The soft margin (equation (6)) is thus quadratic in both the input projection $u$ and the readout vector $v$ and therefore simple to optimize with respect to either of them. Hereby, we require both projection vectors to be normalized. For the readout, this ensures a meaningful calculation of the margin. For the input projection, this fixes the amplitude of the driving signal (cf. section 4 and supplementary material (section A)). A constrained optimization follows with the method of Lagrange multipliers by computing the stationary points of

$$\mathcal{L}(u,v) := \kappa_\eta(u,v) + \lambda_u(\|u\|^2 - 1) + \lambda_v(\|v\|^2 - 1) \qquad (8)$$

with $\lambda_{u/v} < 0$ (see supplementary material, section A.3). This equation can be maximized by alternating fixed-point iteration. In case $\mu = 0$, the soft margin is a quadratic form and finding the optimal projection vectors reduces to an eigenvalue problem. A detailed description of the optimization process is given in the supplementary material, section A.

Figure 2 shows the increase in soft margin by optimizing the input projection $u$ in the linear reservoir. Inspecting the time span just prior to the readout time point $T$ exposes the high sensitivity of the soft margin to the readout time point (Figure 2a). The global optimum may thus be reached at some intermediate time point, prior to the end of the stimulus; it is possible that later steps of the stimuli counteract the separation or disturb the favorable orientation of the state clouds. On average over many sets of stimuli, however, the soft margin increases towards late readout times (Figure 2b), indicating that the reverberating activity of the network can effectively be used to accumulate evidence. Networks closer to instability with longer time constants or a time-integrated readout may therefore be beneficial for the performance, particularly in the example shown in section 5. Qualitatively, the dominance of the more recent past of the stimulus, however, prevails. An average over stimuli of the decomposition $\omega_\alpha = w_\alpha^\mathrm{T} u$ of the optimal input projections into eigenmodes $w_\alpha$ of the connectivity,

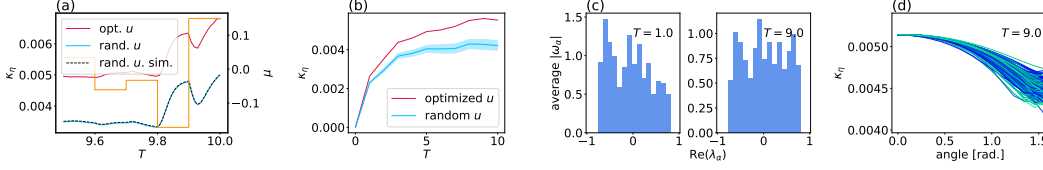

Figure 2: *Optimization of the input and readout projections in a reservoir of linear units.* (a) Random network with $N = 100$ neurons, $\tau = 0.25$ and fixed connectivity $W$ ($W_{ij} \sim \mathcal{N}(0, \frac{g^2}{N})$, $g = 0.9$) is stimulated by step-wise constant stimuli with a mean separation of $\mu = \frac{1}{2}(x_+(t) - x_-(t))$ (orange, right scale). The soft margin (left scale) when stimulated with a random input projection, but readout vector optimized according to equation (6) (blue: analytical solution (equation (3)), black dashed: simulation result [Gewaltig and Diesmann, 2007]) with $\eta = 10$ is shown alongside the combined optimization using 30 optimization steps of input and readout projection at readout time $T$ (red). (b) The same network is stimulated with 250 random stepwise constant signals with optimized (red) and 250 random (blue) input projections each. The corresponding readout projection is chosen optimally in either case. The soft margins, averaged over stimuli, are shown for both cases. Colored area marks the standard deviations of $\kappa_\eta$ with respect to random input projections averaged over stimuli. (c) At readout times $T = 1$ and $T = 9$, the optimal input projections of the samples used in (b) are decomposed into eigenmodes of the reservoir. Histograms show the average absolute weight $\omega_\alpha$ of the modes corresponding to an eigenvalue $\lambda_\alpha$; real part determines the time constant $\tau_\alpha$ of the mode. (d) Soft margin for varying input projection $u$ that has the given angle on the abscissa to the optimal direction $u^*$ at readout time $T = 9$; $u$ is oriented within a randomly chosen hyperplane.

$w_\alpha^{\mathrm{T}} W = \lambda_\alpha w_\alpha^{\mathrm{T}}$, is shown in Figure 2(c). The information projected on each mode thereby decays exponentially with time constant $\tau_\alpha = \tau \left(1 - \mathrm{Re}(\lambda_\alpha)\right)^{-1}$. Pronounced contributions of modes with short time constants at both early and late readout times emphasize the importance of the recent past of the stimulus for classification. Perturbing the input projection $u$ into random directions shows that the optimal direction is sharply defined (Figure 2(d)).

## 4 Non-linear Networks

Classification by a linear system fails when stimuli become linearly inseparable, because the mapping of the stimulus into the state space of the network can only perform a linear transformation. The introduction of a non-linear activation function qualitatively changes this result. Interpreting the processing in the network as a kernel functional, the space it belongs to is thus extended: to leading order in a perturbative expansion, the mapping changes from a linear functional to a quadratic functional; that is, a functional in which pairs of time points of the input signal contribute to the network output at any given point in time. These non-linear interactions render the system sensitive to class-specific characteristics also in higher order cumulants. The soft margin therefore profits from more contributions to the distance $M$ and covariance $\Sigma$ of the state clouds. The approach therefore elucidates which statistical features of the input data can be used by the network, thus opening a door to link and compare reservoir computing to feature-based approaches of classification.

We focus on the case where the neural gain function is explored only in a confined area around a working point, where the non-linearity remains small, so we expand the gain function as $\phi(y) \simeq y + \alpha y^2 + \mathcal{O}(y^3)$ with a small, positive parameter $0 \leq \alpha \ll 1$, and a small or vanishing initial condition for $y$. In the context of biological neural networks, the gain function represents the non-linearity experienced by a single synaptic input on the background noise caused by the other inputs. It is formally ontained by a Gram-Chalier expansion; an expansion in the non-Gaussian cumulants of a nearly Gaussian distributed input. [Dahmen et al., 2016] and [Farkhooi and Stannat, 2017] have explored such expansions for binary networks and found that even the linear order provides a good approximation of the recurrent dynamics, as soon as the number of inputs per neuron is on the order of $50 - 100$. For conceptual clarity, we here focus on the simpler case of a rate network, but more elaborate methods are also conceivable. The corresponding Green's function for the network can be derived from a perturbation expansion of the corresponding network dynamics in orders of $\alpha$ as

$$y(t) = y^{(0)}(t) + \alpha y^{(1)}(t) + \mathcal{O}(\alpha^2). \tag{9}$$

Inserting the ansatz (equation (9)) into equation (1) separates the solution into different orders of $\alpha$. The zeroth order,

$$\left(\tau\partial_t + 1\right) y_i^{(0)}(t) - \sum_j W_{ij} y_j^{(0)}(t) = u_i x(t) + \mathcal{O}(\alpha), \tag{10}$$

recovers the linear system, solved by equation (3). Corrections to the dynamics can be found in higher orders in $\alpha$. With use of equation (10), the differential equation (equation (1)) with terms up to first order in $\alpha$ simplifies as

$$\left(\tau\partial_t + 1\right) y_i^{(1)}(t) - \sum_j W_{ij} y_j^{(1)}(t) = \sum_j W_{ij} (y_j^{(0)}(t))^2. \tag{11}$$

The first non-linear correction to the linear dynamics obeys the same differential equation as the linear one, with the linear solution entering the inhomogeneity in the place of $u_i x(t)$. Thus, $y^{(1)}$ follows with the Green's function $G^{(1)}$ (equation (2)) and equation (11) as

$$\alpha y_i^{(1)}(t) = \alpha \sum_{i',j} \int_{-\infty}^{\infty} \mathrm{d}t'\, G_{ii'}^{(1)}(t,t')\, W_{i'j} \left[ y_j^{(0)}(t') \right]^2$$

$$=: \sum_{p,q} \int_{-\infty}^{\infty} \mathrm{d}s \int_{-\infty}^{\infty} \mathrm{d}s'\, G_{ipq}^{(2)}(t,s,s')\, u_p u_q\, x(s)x(s'), \tag{12}$$

where we defined the second order Green's function $G^{(2)}$ and $y_j^{(0)}(t')$ is the zeroth order solution of equation (10) given by equation (3). At this order, the reservoir thus maps the input by a bi-linear functional kernel to the output. Concerning the validity of the approximation, it must be noted that, whereas the solution of the linear system remains well-defined also in the linearly unstable regime, the perturbative solution of the non-linear system built thereof (equations (9) - (12)) in that case suffers from exponentially growing modes. Therefore, we do not consider chaotic networks in our analysis, restricting the variance of connectivity weights to $g < 1$.

Figure 3(a) shows that the first order correction in $\alpha$ approximates the dynamics of the full system quite well. For small $\alpha$, the network is linearly stable: the eigenvalues $\tilde{\lambda}$ of the linearized connectivity (see supplementary material, section A.2) $\tilde{W}_{ij} = W_{ij}(1 + 2\alpha y_j(t))$ fulfill $\max(\mathrm{Re}(\tilde{\lambda})) < 1$ (Figure 3a, right inset). Consequently, the difference between the linear and non-linear system is not large. Yet, we will show that the non-linearity has a considerable impact on the separability of inputs where the linear theory alone fails to separate the stimuli.

Given the Green's functions $G^{(1)}$ and $G^{(2)}$, the expected distance and covariance required for evaluation of the soft margin in equation (6) can be computed using

$$\zeta_\nu y_i^{u,\nu} = \sum_{p,n} G_{ipn}^{(1)}(T)\, u_p\, \zeta_\nu x_n^\nu + \sum_{p,q,m,n} G_{ipqnm}^{(2)}(T)\, u_p u_q\, \zeta_\nu x_n^\nu x_m^\nu. \tag{13}$$

The distance $M$ between the state clouds thereby receives $\mathcal{O}(\alpha)$ contributions from the first two cumulants of the stimuli, whereas the covariance $\Sigma$ receives corrections up to $\mathcal{O}(\alpha^2)$ from stimulus cumulants up to fourth order. Although $\mathcal{O}(\alpha^2)$ corrections to $\Sigma$ form only a small modification to the covariance that is otherwise determined up to $\mathcal{O}(\alpha)$, this term is essential to guarantee its positive definiteness. A consistent calculation of network state cumulants is therefore required for a stable optimization algorithm. It is easy to show that all orders in $\alpha$ of both $M$ and $\Sigma$ are affected by Gaussian distributed stimuli. The latter is therefore the minimal example to expose cumulant-mixing based on non-linearities. Contributions from higher order cumulants of stimuli would not show qualitatively different effects.

Equation (8) can thus be expressed with help of equation (13). As in the linear case it is bi-linear in $v$, but due to $\Sigma$ it now contains terms with third and fourth power in $u$. By the bi-linearity in $v$, the readout projection is determined as in the linear case, only with additional contributions to the covariance matrix and distance vector. The optimization of the input projection, by contrast, is more challenging. The higher powers of $u$ impede a direct solution. In our analysis, the most reliable optimization scheme proved to be searching for a direct solution to $\partial_u \mathcal{L}(u,v) = 0$ given by equation (8) with an appropriate initial guess. More details and pseudocode can be found in the supplementary material (section A.3).

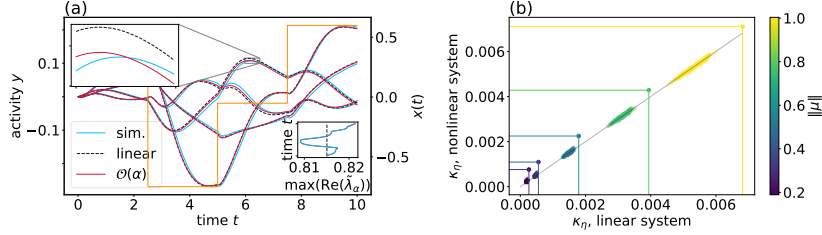

Figure 3: *Responses and soft margins in a network with small non-linearity $\alpha$.* (a) First order approximation $\mathcal{O}(\alpha)$ of the dynamics (red) and linear response (black dashed). Simulation shown in blue. Left inset shows a zoom in, right inset shows time evolution of $\max_\alpha(\text{Re}(\tilde{\lambda}_\alpha))$ for simulation (blue) and $\mathcal{O}(\alpha)$ approximation (red) of the nonlinear reservoir together with $\max_\alpha(\text{Re}(\lambda_\alpha))$ (black dashed) of the linear reservoir. (b) Soft margins $\kappa_\eta$ for random (stars) and optimized input projections $u$ (dots) for one example network realization; both cases use optimized readout projection $v$ with respect to equation (8). Linear system on x-axis, non-linear system on y-axis. Vertical and horizontal colored lines at position of optimized solution provided as a guide. Gray line is the angle bisector. Colors indicate $\|\mu\|$, the strength of linear separability of the underlying stimulus distribution, where $\|\mu\| \in \{0.19, 0.30, 0.52, 0.76, 1.0\}$ from violet to yellow. Both $\psi$ and $\chi$ are held constant with eigenvalues of $\psi \pm \chi$ in the range $[0.3, 2.2]$. Same parameters as in figure 2, but with $\alpha = 0.05$ for the non-linear system.

How much the choice of the input projection affects the soft margin can be observed in figure 3(b). The input and readout projections are optimized separately for a linear and a non-linear network; they take on different optimal values for the two reservoirs. A benefit of optimizing the input projection in the linear reservoir only occurs for increasing strength in the mean class difference $\mu$. For small $\mu$, the optimal direction is dominated by the one that minimizes the effect of the noise, while for larger $\mu$, the stimulus direction aligns such as to maximize the mean separation of the output of the network. The situation is clearly different in the non-linear reservoir even for the weak non-linearity considered here: At low linear separabilities of inputs, the optimization of the input projection in the non-linear reservoir yields a strong relative improvement in separability of outputs, indicated by the soft margin. For linearly well separable classes the relative improvement with respect to the linear reservoir shrinks, while the absolute improvement stays rather constant. Close to the information theoretic optimum of perfectly separable classes considered in the dataset application in section 5, the benefit of non-linearities becomes negligible. The superior performance of the weakly non-linear system with respect to the linear system vanishes for all input separabilities if input projections are not optimized: For random input projections, the performance in the non-linear reservoir is on average only slightly better, and sometimes even worse, than in the linear reservoir. The random input projections accumulate along the identity line in figure 3(b), with a center of mass slightly in the upper area.

In summary, while the weak non-linear corrections to the linear dynamics as used here do not exploit the full computational power non-linearities can exert, the presented routine allows us to inspect the potential of this framework that is not apparent in classical reservoir computing with random input projections.

## 5  Application to ECG5000 dataset

We conclude the analysis with an application to a univariate temporal classification dataset. This serves as a proof-of-concept to demonstrate the effects of the optimization on a real-world problem and can be regarded as a check that real data do not generally contain structural obstacles that were not covered in the theoretical considerations. To raise the method from the proof-of-concept level, the performance should be systematically checked on a broader set of problems as done for state of the art time series classifiers [Bagnall et al., 2017, Wang et al., 2017], which we leave for future work.

We here restrict the preprocessing of the data to a minimum. In this spirit, also the free parameters $\eta$ and $\tau$ are chosen appropriately, but not optimally. The focus lies solely on a comparison between random and optimized input projections. This comparison is based on the classification soft margin

Table 1: *Quality measures for the application of the optimization scheme to ECG5000.* Soft margin $\kappa_\eta$ (left) and accuracy (right) for optimized and 50 random input projections, averaged over 20 different network realizations.

|  | $\kappa_\eta$, linear | $\kappa_\eta$, non-linear | accuracy, linear | accuracy, non-linear |
|---|---|---|---|---|
| random $u$ | $0.182 \pm 0.015$ | $0.183 \pm 0.015$ | $(91.7 \pm 0.6)\,\%$ | $(91.7 \pm 0.7)\,\%$ |
| optimized $u$ | $0.383 \pm 0.026$ | $0.384 \pm 0.026$ | $(97.3 \pm 0.4)\,\%$ | $(97,3 \pm 0.4)\,\%$ |

and accuracy in a fixed reservoir configuration. We can then observe the effect of the optimization routine on the separation and covariance of the state clouds.

The examined dataset is ECG5000, which is publicly available at the UCR Time Series Classification archive [Chen et al., 2015], containing 5000 electrocardiograms of single heartbeat recordings. The classes separate between five categories of healthy and diseased heartbeats. For a binary classification, we use only samples from the two largest classes, so that we obtained a training set consisting of 354 samples and a testing set of 4332 samples. All stimuli were shifted and scaled to provide classes with means $\pm\mu$ with $\|\mu\| = 1$; higher order cumulants changed accordingly. This scaling of inputs is only performed for conceptual clarity, allowing identical network parameters as in the previous task. Likewise, one could adapt the value of $\alpha$ according to the stimulus strength. Furthermore, for maximal performance, a trained threshold can replace the centering of data. As a measure of linear separability, we relate the difference of the class means to the covariance in the direction of separation. This yields a ratio $\|\mu\|^2/\sqrt{\mu^T \psi \mu} = 2.6$, which is much higher than for the artificial stimuli analyzed in figure 3, where the corresponding measure ranges between 0.19 and 0.98.

All results presented here use the same parameters as in figure 2 and figure 3. The summary of the results in table 1, which contains averaged results over 20 different initializations of the recurrent connectivity, makes evident that a maximized soft margin is accompanied by increased accuracies. The optimized input projections outperformed all randomly chosen ones both with respect to soft margin and accuracy. Because of the close to perfect linear separability of the data, the increase of soft margins and accuracies from the linear to the non-linear reservoir is very small (see supplementary material, section A.7). These results are as theoretically expected from figure 3(b) for linearly well separable data. An application to a broader set of real world data would be required to quantify the performance increase in terms of accuracy also in the case of linearly less separable stimuli.

A visualization of the optimization in figure 4 shows the increase of distance between the classes before and after optimization (a, b), gradually increasing with the optimization step (c). The projections being optimized for $T = 10$, the two classes become distinguishable only shortly before this readout time point (d). The variance along the readout direction $\frac{\eta}{2} v^T \Sigma^u v$ is hereby rather constant, while the main deviations occur due to the separation. Increasing $\eta$ can be used to enforce smaller dispersion of the state clouds (see supplementary material, section A.7). The enlarged range in between the class centers with low probability density for network states of either class facilitates a better generalization to unknown data.

## 6 Discussion

We present an analytical approach of unrolling recurrent non-linear networks by use of a perturbative expansion. The conceptual insight of this step lies in a simplification of the reverberating neuronal dynamics into an effective feed-forward structure. This approach, which involves the first and second order Green's function of the system, extends naturally from linear networks to non-linear ones. The reformulation of the classification margin as a partly concave soft margin, which has a similar form as a free energy, facilitates the derivation of closed-form expressions to be maximized. The joint optimization of stimulus projection and readout vector leads to a significant increase in classification performance by tuning the network state distribution towards a trade-off between low variability along the direction of separation and high absolute separation. This increase of separability is in particular observable even in only weakly non-linear networks when the linear separability of the stimuli is low. The effect can be fully explained by the second order Green's function that makes the reservoir sensitive to classification features in the second order stimulus statistics. We find that the effect of higher statistical orders of the data are suppressed by powers in the perturbation parameter, the non-linearity of the neuronal dynamics. But also the classification performance of the linearly

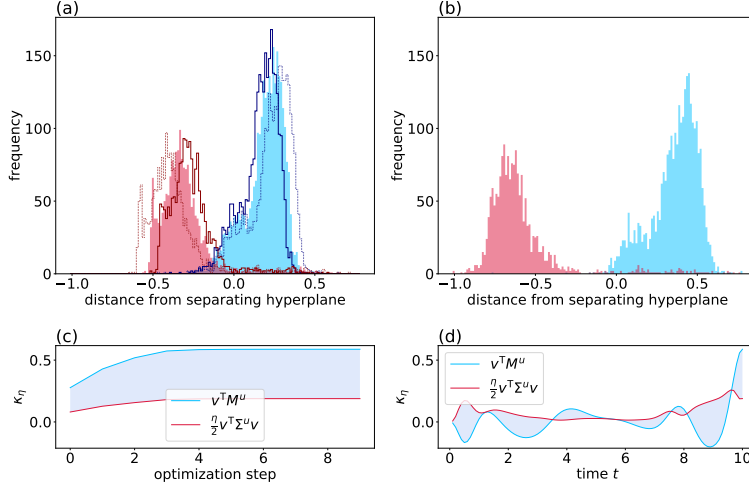

Figure 4: *Network state distribution for random and optimal input projection.* (a) Random input projections: Colored histogram shows the network state distribution for both classes in direction of the separating hyperplane's normal vector (average over 50 input projections). Light and dark outlined histograms correspond to the input vectors with the best and worst accuracies among the drawn samples, respectively. All network states are based on $\mathcal{O}(\alpha)$-predictions of the dynamics. (b) Optimized input projection: Histogram of the network state distribution (based on the simulated responses). (c) Evolution of distance and covariance contribution to the soft margin $\kappa_\eta$ over the first 10 optimization steps at readout time. Height of the shaded area corresponds to the resulting $\kappa_\eta$, illustrating the difference between the two terms in equation (6). (d) Evolution of distance and covariance contribution to $\kappa_\eta$ over simulation time for optimized input vector $u$. Height of the shaded area corresponds to the soft margin (negative where covariance contribution (red) exceeds distance contribution (blue) and positive otherwise). Same network and parameters as in figure 3.

well separable dataset ECG5000 profits significantly from the optimization. The framework presents a stepping stone towards a systematic understanding of information processing by recurrent random networks.

## Acknowledgements

This work was partly supported by European Union Horizon 2020 grant 945539 (Human Brain Project SGA3), the Helmholtz Association Initiative and Networking Fund under project number SO-092 (Advanced Computing Architectures, ACA), BMBF Grant 01IS19077A (Juelich) and the Excellence Initiative of the German federal and state governments (G:(DE-82)EXS-PF-JARA-SDS005, G:(DE-82)EXS-SF-neuroIC002).

## Broader impact

The main motivation of this work is to provide conceptual insight. Analytically unrolling recurrent dynamics into a (functional) Taylor series, where coefficients are given by Green's functions, is a versatile approach that may be used as a general purpose scheme to analyze recurrent networks and to optimize reservoir computing. This expansion reveals how the non-linear interactions and recurrence pick up higher order correlations in the input statistics, quantifying how non-linear networks provide a richer feature space than linear ones. We consider the presented application as a proof-of-principle for optimized processing of complex time series data. The presented application to health-related data (heartbeat classification) hints at possible societal consequences by providing better diagnostic tools.

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
