[Supplementary Material 1 · Appendix.pdf]

# A  Supplementary material

## A.1  Convexity of the soft-margin

The soft margin $\kappa_\eta$ (equation (5)) is concave in $v$; this follows directly from the linear appearance of $v$ in the exponent of the exponential function with Hoelder's inequality. Hoelder's inequality states for two non-negative sequences $g_k, h_k \geq 0$ and for $\alpha + \beta = 1$ that

$$\sum_k (g_k)^\alpha (h_k)^\beta \leq \left( \sum_k g_k \right)^\alpha \left( \sum_k h_k \right)^\beta. \tag{14}$$

We here follow a modified version of the argument in [Goldenfield, 1992]. It therefore follows for $\alpha + \beta = 1$ that

$$\kappa_\eta(u, \alpha v_1 + \beta v_2) = -\frac{1}{\eta} \ln \sum_{\nu=1}^{P} \exp\left( \alpha \left[ -\eta \, \zeta_\nu \left( v_1^{\mathrm{T}} y^{u,\nu} \right) \right] + \beta \left[ -\eta \, \zeta_\nu \left( v_2^{\mathrm{T}} y^{u,\nu} \right) \right] \right)$$

$$= -\frac{1}{\eta} \ln \sum_{\nu=1}^{P} \exp\left( -\eta \, \zeta_\nu \left( v_1^{\mathrm{T}} y^{u,\nu} \right) \right)^\alpha \exp\left( -\eta \, \zeta_\nu \left( v_2^{\mathrm{T}} y^{u,\nu} \right) \right)^\beta$$

$$\overset{\text{Hoelder}}{\geq} -\frac{1}{\eta} \ln \left[ \sum_{\nu=1}^{P} \exp\left( -\eta \, \zeta_\nu \left( v_1^{\mathrm{T}} y^{u,\nu} \right) \right) \right]^\alpha \left[ \sum_{\nu=1}^{P} \exp\left( -\eta \, \zeta_\nu \left( v_2^{\mathrm{T}} y^{u,\nu} \right) \right) \right]^\beta$$

$$= \alpha \kappa_\eta(u, v_1) + \beta \kappa_\eta(u, v_2).$$

## A.2  Linearized connectivity

The effective connectivity $\tilde{W}$ is obtained from linearizing around the network's time evolution as

$$(\tau \partial_t + 1)(y_i(t) + \delta y_i(t)) = \sum_j W_{ij}((y_j(t) + \delta y_j(t)) + \alpha(y_j(t) + \delta y_j(t))^2) + u_i x(t)$$

$$\Rightarrow (\tau \partial_t + 1)\delta y_i(t) = \sum_j W_{ij}(1 + 2\alpha y_j(t)) \, \delta y_j(t) + \mathcal{O}(\delta y^2)$$

and approximating the non-linear system by an equivalent linear one with connectivity $\tilde{W}_{ij} = W_{ij}(1 + 2\alpha y_j(t))$. The evolution of the system becomes unstable when the real part of an eigenvalue $\tilde{\lambda}_\alpha$ of the effective matrix $\tilde{W}$ exceeds 1. The time evolution of $\max_\alpha(\mathrm{Re}(\tilde{\lambda}_\alpha))$ displayed in Figure 3(a) for the full system and the $\mathcal{O}(\alpha)$ approximation assures the stability of the solution and the quality of the approximation.

## A.3  Constrained optimization with Lagrange multipliers

We need to optimize equation (8)

$$\mathcal{L}(u, v) := \kappa_\eta(u, v) + \lambda_u(\|u\|^2 - 1) + \lambda_v(\|v\|^2 - 1),$$

where $\kappa_\eta$ takes the form of equation (6). Although the mathematical structure of equation (6) is simple, the optimization of the expression may present a few pitfalls. In this section, we describe in detail how to find the projection vectors given the first four moments of the stimuli.

The linear system can be understood as a special case of the non-linear system, where some contributions to the soft margin and its gradients vanish. Therefore, we will distinguish the types of reservoir kernels only where they are relevant.

## A.4  Prerequisites

The numerical results of the optimization slightly depend on the value of the control parameter $\eta$ of the soft margin that has to be fixed. In our examples, with $\eta = 10$ the soft margin showed already very similar extrema as the margin. Smaller values correspond to softer margins. In practice, a good choice of $\eta$ can be obtained by comparing for different $\eta$ the optimized readout vector and accuracies

for responses of some reservoir to an arbitrary stimulation. This procedure is fast and reliable since finding the readout vector for some $\eta$ is only a quadratic problem. Furthermore, an analysis of the time evolution of the soft margin for random input projections can be used as in figure 2(b) to achieve a good estimate of a suitable $\tau$. It can be chosen such that the soft margin for random stimuli just entered a saturating phase, so that there is not much improvement expected. Extended phases of saturation, however, are a sign of forgetting of early parts of the stimuli in the network and should be avoided.

The main procedure then consists of an alternating optimization of the input and readout projections. Thereby, we denote $\mathcal{L}(u|v)$ as the objective function for input optimization, given $v$, and $\mathcal{L}(v|u)$ analogously.

### A.5 Optimization of the input projection

The determination of the input projection for fixed readout vector is best conducted, depending on the situation, by one of three methods for non-linear kernels and one of two methods for linear ones. The quantities

$$m_{0p} = \sum_{i,n} G^{(1)}_{ipn} v_i \langle \zeta_\nu x^\nu_n \rangle$$

$$m_{1pq} = \sum_{i,n,m} G^{(2)}_{ipqnm} v_i \langle \zeta_\nu x^\nu_n x^\nu_m \rangle$$

$$\sigma_{0pq} = \sum_{\substack{i,j, \\ n,m}} \frac{1}{2}\eta\, G^{(1)}_{ipn} G^{(1)}_{jqm} v_i v_j (\langle x^\nu_n x^\nu_m \rangle - \langle \zeta_\nu x^\nu_n \rangle \langle \zeta_\nu x^\nu_m \rangle)$$

$$\sigma_{1pqr} = \sum_{\substack{i,j, \\ n,m,o}} \frac{1}{2}\eta\, (G^{(1)}_{ipn} G^{(2)}_{jqrmo} + G^{(2)}_{iqrmo} G^{(1)}_{jpn}) v_i v_j (\langle x^\nu_n x^\nu_m x^\nu_o \rangle - \langle \zeta_\nu x^\nu_n \rangle \langle \zeta_\nu x^\nu_m x^\nu_o \rangle)$$

$$\sigma_{2pqrs} = \sum_{\substack{i,j, \\ n,m,o,l}} \frac{1}{2}\eta\, G^{(2)}_{ipqmn} G^{(2)}_{jrsol} v_i v_j (\langle x^\nu_n x^\nu_m x^\nu_o x^\nu_l \rangle - \langle \zeta_\nu x^\nu_n x^\nu_m \rangle \langle \zeta_\nu x^\nu_o x^\nu_l \rangle),$$

where $G^{(2)}$ is the $\mathcal{O}(\alpha)$ correction of the Green's function $G^{(1)}$ for linear kernels and $\mu = \langle \zeta_\nu x^\nu_n \rangle$, $\psi = \langle x^\nu_n x^\nu_m \rangle - \langle \zeta_\nu x^\nu_n \rangle \langle \zeta_\nu x^\nu_m \rangle$ and $\chi = \langle \zeta_\nu x^\nu_n x^\nu_m \rangle$ constitute the cumulants in the notation (equation (7)) used in the main text. This notation is introduced here and in section A.6 for legibility, although a memory-efficient implementation will compute only products of Green's functions with stimuli $x^\nu$ (for example, $\mathcal{G}^{(2)}_{ipq\nu} = \sum_{n,m} G^{(2)}_{ipqnm} x^\nu_n x^\nu_m$), which then compose the above abbreviations by performing the averages over $\nu$. With these abbreviations, the dependence of the soft margin on the input projection reads

$$\kappa_\eta = \sum_p u_p m_{0p} + \sum_{p,q} u_p u_q m_{1pq} - \sum_{p,q} u_p u_q \sigma_{0pq} - \sum_{p,q,r} u_p u_q u_r \sigma_{1pqr} - \sum_{p,q,r,s} u_p u_q u_r u_s \sigma_{2pqrs}.$$

**Preparations for optimization in non-linear systems**

In the non-linear case, it is advisable to take a few precautions to reduce computation time and enhance performance. The determination of the optimal input projection $u$ given a fixed readout projection $v$ should in the first few, but at least one, iterations neglect terms of $\mathcal{O}(\alpha)$ and higher in the covariance $\Sigma^u$. In these steps, the soft margin is not strictly optimized, but the result still yields a good initial guess for the full problem. The advantage of this procedure is that the computation is much faster and more likely to achieve a solution near the optimum rather than some local extremum. In the first steps, the direction of the projection vector $u$ typically changes rapidly and the quadratic part alone often has a maximum near the optimum of the full soft margin, as the neglected terms are at least $\mathcal{O}(\alpha)$. In the readout optimization, the problem is in general quadratic in case of both linear and non-linear dynamics, so there is no need to make further simplifications.

Furthermore, the soft margin is not necessarily convex in $u$ in the non-linear case and sometimes exhibits plateaus over iteration steps. We therefore recommend to use a small number of initial projection vectors, optimize them over a few steps as described below, and then proceed with the best one after these steps.

**Case A.**

The simplest case arises if $\|\mu\| = 0$, since then $m_0$, $\sigma_1$ and $\sigma_2$ vanish, if one neglects the $\mathcal{O}(\alpha)$ contributions in the non-linear case as discussed above. For normalized input projections, equation (6) is then maximized by the eigenvector corresponding to the smallest eigenvalue of $\sigma_0 - m_1$.

**Case B.**

In the general case where $\|\mu\| \neq 0$, it is, as mentioned before, sometimes helpful to ignore the part related to $\sigma_1$ and $\sigma_2$ of the soft margin in the non-linear case to obtain a good guess of the input projection that maximizes equation (8). Since $m_1$, $\sigma_1$ and $\sigma_2$ vanish when $\alpha = 0$, the same procedure applies in the linear case. The objective then reads

$$\mathcal{L}(u|v) \to u^{\mathrm{T}} m_0 + u^{\mathrm{T}} m_1 u - u^{\mathrm{T}} \sigma_0 u + \lambda_u (u^{\mathrm{T}} u - 1),$$

so $u$ and $\lambda_u$ are found using

$$\partial_u \mathcal{L} = 0 \Rightarrow 2(\sigma_0 - m_1 - \lambda_u \mathbb{I}) u = m_0, \tag{15}$$

$$\partial_{\lambda_u} \mathcal{L} = 0 \Rightarrow u^{\mathrm{T}} u - 1 = 0. \tag{16}$$

These equations have many solutions, but for a maximum we further require negative definiteness of $\partial_u^2 \mathcal{L}|_{\lambda_u}$. From this condition follows that $\lambda_u < \min\{\sigma \,|\, \sigma \text{ is eigenvalue of } \sigma_0 - m_1\}$. Then, $\sigma_0 - m_1 - \lambda_u \mathbb{I}$ is symmetric and invertible and, from solving the first condition for $u$ and inserting in the second, we get

$$\frac{1}{4}(m_0^{\mathrm{T}} (\sigma_0 - m_1 - \lambda_u \mathbb{I})^{-1} (\sigma_0 - m_1 - \lambda_u \mathbb{I})^{-1} m_0) = 1. \tag{17}$$

The term on the left hand side is positive, has poles around the eigenvalues of $\sigma_0 - m_1$ and deviates only slightly from 0 for $\lambda_u \ll \min\{\sigma \,|\, \sigma \text{ is eigenvalue of } \sigma_0 - m_1\}$. A bisection is therefore best suited to determine $\lambda_u$ and thereby $u$ using equation (15). However, the poles have only a very small width and the determination of eigenvalues and inverse matrices is accompanied by numerical uncertainties. Therefore, the upper bound on $\lambda_u$ is found best as the smallest value within a window of a small width $\varepsilon$ around the smallest eigenvalue, where the term on the left hand side exceeds one. Although this corresponds to a fine-tuning of the Lagrange parameter $\lambda_u$ with a sensitive dependence of the left hand term in equation (17) on the exact used eigenvalues, the soft margins corresponding to the obtained solutions remained robust against neglecting near-vanishing, and therefore numerically uncertain, eigenvalues in the summation. Components of the input projection in these directions are neutralized by their eigenvalues in equation (8).

**Case C.**

If the system is non-linear and a good initial guess for the input projection is available, predefined solvers, such as the fsolve function implemented in numpy [Oliphant, 2006], typically find good solutions for the Lagrange conditions, which are in this case

$$2(\sigma_{0pq} - m_{1pq} - \lambda_u \mathbb{I}_{pq}) u_q + (\sigma_{1pqr} + \sigma_{1qrp} + \sigma_{1rpq}) u_q u_r$$
$$+ (\sigma_{2pqrs} + \sigma_{2qrsp} + \sigma_{2rspq} + \sigma_{2spqr}) u_q u_r u_s = m_{0p},$$
$$u^{\mathrm{T}} u = 1.$$

The first guess should be the solution from the previous iteration step. Only if the soft margin reduces by the found solution, a new guess should be computed neglecting $\sigma_1$ and $\sigma_2$. For this comparison, it is important to make sure the projection vectors are properly normalized. Although this is ensured by the Lagrange condition, the actual lengths of the returned vectors slightly deviate from one because of the fine-tuning of the Lagrange parameters. If the soft margin found near that solution still decreases, we decided to use the new solution anyway as a restart-point. The readout vector optimization then improves the soft margin again.

## A.6 Optimization of the readout projection

The optimization of the readout projection is structurally the same as for the input projection, only the objective function is in general bi-linear in $v$. The abbreviations used here are

$$M_{0i} = \sum_{p,n} G^{(1)}_{ipn} u_p \langle \zeta_\nu x_n^\nu \rangle$$

$$M_{1i} = \sum_{\substack{p,q,\\n,m}} G^{(2)}_{ipqnm} u_p u_q \langle \zeta_\nu x_n^\nu x_m^\nu \rangle$$

$$\Sigma_{0ij} = \sum_{\substack{p,q,\\n,m}} G^{(1)}_{ipn} G^{(1)}_{jqm} u_p u_q (\langle x_n^\nu x_m^\nu \rangle - \langle \zeta_\nu x_n^\nu \rangle \langle \zeta_\nu x_m^\nu \rangle)$$

$$\Sigma_{1ij} = \sum_{\substack{p,q,r,\\n,m,o}} (G^{(1)}_{ipn} G^{(2)}_{jqrmo} + G^{(2)}_{iqrmo} G^{(1)}_{jpn}) u_p u_q u_r (\langle x_n^\nu x_m^\nu x_o^\nu \rangle - \langle \zeta_\nu x_n^\nu \rangle \langle \zeta_\nu x_m^\nu x_o^\nu \rangle)$$

$$+ \sum_{\substack{p,q,r,s,\\n,m,o,l}} G^{(2)}_{ipqmn} G^{(2)}_{jrsol} u_p u_q u_r u_s (\langle x_n^\nu x_m^\nu x_o^\nu x_l^\nu \rangle - \langle \zeta_\nu x_n^\nu x_m^\nu \rangle \langle \zeta_\nu x_o^\nu x_l^\nu \rangle).$$

The objective function to maximize is then

$$\mathcal{L}(v|u) \to v^{\mathrm{T}}(M_0 + M_1) - \frac{1}{2}\eta v^{\mathrm{T}}(\Sigma_0 + \Sigma_1)v + \lambda_v(v^{\mathrm{T}}v - 1).$$

Only if the system is linear and the mean stimulus difference $\mu$ is vanishing, this becomes an eigenvalue problem and the optimal readout vector $v$ is the eigenvector corresponding to the smallest eigenvalue of $\Sigma_0$ (compare case A). Otherwise, the Lagrange parameter follows from a bisection using the conditions

$$\partial_v \mathcal{L} = 0 \Rightarrow (\eta(\Sigma_0 + \Sigma_1) - 2\lambda_v \mathbb{I})v = M_0 + M_1, \tag{18}$$

$$\partial_{\lambda_v} \mathcal{L} = 0 \Rightarrow v^{\mathrm{T}}v - 1 = 0. \tag{19}$$

From negative definiteness, $\lambda_v < \frac{1}{2} \min\{\sigma \,|\, \sigma \text{ is eigenvalue of } \eta(\Sigma_0 + \Sigma_1)\}$ follows as upper bound on $\lambda_v$ (compare case B).

## A.7 Additional material

A folder with example figures for different network realizations as in figure 3 can be found in the supplementary material folder (data/responses_soft_margins). The folder data/ECG contains some of the softmargins and accuracies of the reservoirs used to generate table 1. The relation between linear and non-linear optimal softmargins and accuracies can be evaluated with this data, as well as an analysis presented in figure figure 4 for different realizations of the connectivity. Data for the same networks, but optimized with $\eta = 30$, can be found in the folder data/ECG/additional. As can be verified with this data, the equivalent of table 1 for this case, averaged over 10 network realizations, reads

|  | $\kappa_\eta$, linear | $\kappa_\eta$, non-linear | accuracy, linear | accuracy, non-linear |
|---|---|---|---|---|
| random $u$ | $0.109 \pm 0.010$ | $0.109 \pm 0.010$ | $(94.6 \pm 0.7)\%$ | $(94.7 \pm 0.7)\%$ |
| optimized $u$ | $0.196 \pm 0.006$ | $0.196 \pm 0.006$ | $(98.1 \pm 0.1)\%$ | $(98.1 \pm 0.1)\%$ |

Furthermore, the README.md contains instructions on how to set up the environment to reproduce, modify and utilize the optimizations performed in this work.



[Supplementary Material 2 · fig2_1005.pdf]



(a) Plot with legend: opt. $u$ (red), rand. $u$ (cyan), rand. $u$. sim. (black dashed). Axes: $\kappa_\eta$ (left, values 0.004, 0.005, 0.006), $\mu$ (right, values -0.1, 0.0, 0.1), $T$ (horizontal, values 9.6, 9.8, 10.0). Orange curve shown.

(b) Plot: $\kappa_\eta$ vs $T$. Legend: optimized $u$ (red), random $u$ (cyan). Horizontal axis $T$ (0, 5, 10), vertical axis $\kappa_\eta$ (0.000, 0.002, 0.004).

(c) Two histograms: average $|\omega_\alpha|$ vs $Re(\lambda_\alpha)$. Left: $T = 1.0$ (axis 0.0 to 1.5). Right: $T = 9.0$ (axis 0.00 to 1.00). Horizontal axis $Re(\lambda_\alpha)$ (-1, 0, 1).

(d) Plot: $\kappa_\eta$ vs angle [rad.]. $T = 9.0$. Horizontal axis angle [rad.] (0.0, 0.5, 1.0, 1.5), vertical axis $\kappa_\eta$ (0.0040, 0.0045, 0.0050).

[Supplementary Material 3 · fig3_1001.pdf]



(a) Plot showing activity $y$ versus time $t$, with curves labeled sim. (cyan), linear (black dashed), and $\mathcal{O}(\alpha)$ (red). Inset plots show $x(t)$ versus time $t$ and time $t$ versus $\max(\text{Re}(\tilde{\lambda}_\alpha))$.

(b) Scatter plot of $\kappa_\eta$, nonlinear system versus $\kappa_\eta$, linear system, colored by $\|\mu\|$.

[Supplementary Material 4]



(a) Panel showing activity $y$ and $x(t)$ versus time $t$, with legend entries: sim., linear, $\mathcal{O}(\alpha)$. Inset shows time $t$ versus $\max(\text{Re}(\tilde{\lambda}_\alpha))$.

(b) Scatter plot of $\kappa_\eta$, nonlinear system versus $\kappa_\eta$, linear system, colored by $\|\mu\|$.

[Supplementary Material 5]



(a) Legend: sim., linear, $\mathcal{O}(\alpha)$. Axes: activity $y$, $x(t)$, time $t$. Inset: time $t$ vs $\max(\text{Re}(\tilde{\lambda}_\alpha))$.

(b) Axes: $\kappa_\eta$, nonlinear system vs $\kappa_\eta$, linear system. Colorbar: $\|\mu\|$.