[Reviews · NeurIPS 2020]

Review 1

Summary and Contributions: new analysis of reservoir model through soft margin with interpretation in terms of cumulants and second order Green function

Strengths: - interesting new theoretical approach - empirical evaluation only on one example - good relevance to NeurIPS

Weaknesses: - see below

Correctness: - the title is too general and should be more specific

Clarity: - overall well written

Relation to Prior Work: - could be improved

Reproducibility: Yes

Additional Feedback: The authors develop an interesting analysis, especially with eqs (4)(5)(6) and the second order Green function (12)(13). - the authors study only one example in one application while the title makes a very general claim. Either the authors should demonstrate and compare on many more data sets or should make the title more specific. - interplay recurrent networks and nonlinearities: additional work in the area of recurrent neural networks and nonlinear systems analysis can be mentioned here. - eq (6) is there a connection with Fisher discriminant analysis methods, which are expressed in terms of between and within covariance matrices? - is the choice of the initial state important or not? (probably more important in the nonlinear case?) - are there implicit assumptions on the stability of the system? is global asymptotic stability assumed or can it also be e.g. chaotic? - Table 1: the accuracies in the last two columns are the same, while in the paper it was mentioned that the nonlinear case is much better than the linear one. Please explain. - section 5: unfortunately one example is given here. Can't the method be proposed as a general purpose scheme in different applications? Thanks to authors for the replies, I have taken it into consideration.


Review 2

Summary and Contributions: In this work, the authors study a classification task of one-dimensional temporal sequences. They show that a recurrent neural network (RNN) can act as a spatiotemporal kernel and facilitates the classification of inputs. To unfold the temporal evolution of the recurrent network internal dynamics, they expand the Green function of the network's response to leading orders. By averaging over the input statistics, they arrive at a series expansion in the cumulants of the output. They show that for a randomly connected RNN (a reservoir), the output's classification performance is optimal when the input projections are optimized. They derive a perturbative theory for linear networks and networks with "weak" nonlinearity. Finally, they test their result using labeled time-series data.

Strengths: * The framework of expanding the response of a recurrent network using the Green function is novel and holds great potential. Following ideas from Statistical Mechanic and many-body physics, this method allows a perturbative expansion of complex interaction using a power series in the degree of interaction. While the current work studies a relatively simple case, I think that the work serves as a proof-of-concept that perturbation analysis of the interaction is useful in practical computational problems. * The authors show that the input projections into a recurrent neural network can have a significant impact on the readout performance, at least for a simple classification task. This perspective is different from the bulk of studies on computation by RNNs, which are inspired by reservoir computing with fixed input weights and learned readout. * The paper is written in a clear way accessible to readers without prior knowledge of perturbative methods in many-body physics. * The theoretical predictions are backed up with simulations with artificial inputs and inputs from a known time-series database.

Weaknesses: * The authors address the nonlinearity by expanding the activity around a steady-state, which is a fair assumption. However, the authors further assume that the nonlinear terms, and in particular quadratic terms are small by letting alpha be small in eq (9). It is a strong assumption, which I am not sure is valid in many cases of interest. The authors cite [Roxin et al., 2011], who studied the long-tailed distribution of synaptic input to cortical circuits. The authors state that in vivo, neurons typically operate in a low-med regime. The most common biological models of single neural dynamics (e.g., LIF and HH) include rectification at the transfer function at the origin. In that case, there is a sharp nonlinearity, and the assumption of low alpha does not seem to be valid. Furthermore, one would expect complex computations require operation far from any locally-linear regime, and that networks take advantage of nonlinearities such as rectification. I think the result of nonlinearity's effect should be understood qualitatively and should not be directly compared to cortical circuits. Moreover, the results in figure 3 clearly show that the nonlinearity has a minor contribution to the overall dynamics. * The authors show that classification at time $T$ is mostly affected by the input a short time before it [lines 133-135]. It is a trivial statement since the reservoir network operates in its fixed point regime, and the effect of any input decays exponentially. This point is demonstrated by the Lyapunov spectra analysis of the authors. It is not clear to me if the authors suggest that optimizing the input would result in any other behavior, and if they do, then I don't see any evidence for that. * The result that optimizing the input in nonlinear networks is more significant than linear is somewhat trivial. The optimization of the input projections is equivalent to selecting a feature set for the kernel. For nonlinear networks, the space of features (different spatiotemporal correlations) is much larger. Thus, input optimization — or choosing the best features — will necessarily have a more substantial effect.

Correctness: The method is novel and seems correct. I found some issued with the underlying assumptions (see above)/

Clarity: The paper is written in a clear way accessible to readers without prior knowledge of perturbative methods in many-body physics.

Relation to Prior Work: Yes

Reproducibility: Yes

Additional Feedback: Other minor comments: * In the application to the ECG datasets. The authors normalize the input so that mu=(+/-)1 [line 232]. It is unclear to me how this is done without the knowledge of the labels (perhaps I am missing something). If the data is indeed normalized using the labels, then I see no point in using a real data set, and this test is not preferable to artificial data. * Figure 3a, the right inset has no scale, so it is hard to determine if the correction due to the nonlinear terms is meaningful. I suspect it is not (following my comment from above). * The loss in eq (8) is maximized, so I think the constraints should have a negative sign. The constraints seem remnants of the delta function representation in the partition function. When defining a cost function, I think that either the constraint terms should be squared to normalize the vector, or that they should be interpreted as minimizing the norm (which means having a minus sign, and no need for the "1" term). ****** I don't the authors properly addressed my comments, this may be to limited space in the rebuttal, and the initial high score. While I thinks there are flaws in the study which I hope will be addressed, I stand by my original view that the this work present a new conceptual way to treat the dynamics on RNN and to approach the structure-activity relation.


Review 3

Summary and Contributions: This paper built optimal timeseries classifier upon random reservoir networks. Reservoir network is a kind of random recurrent network which plays a role of the temporal kernel. By developing an analytical approach of unrolling recurrent non-linear networks by use of perturbative expansion, the input projection u and readout vector v can be jointly optimized in this work. A binary timeseries classification task is selected to test the proposed model. Results show that this joint optimization of u and v can lead to significant performance improvement.

Strengths: The derivations in terms of the input projection u and the readout vector v can be calculated in the closed forms.

Weaknesses: More recent related work about recurrent kernel/temporal kernel can be considered. It is unclear what is the advantages of random recurrent networks compared to BPTT training based RNN/LSTMs from the deep learning community. Since reservoir network such as echo state network is much suitable to chaotic time series prediction, when doing timeseries classification tasks, are there any assumptions about the input data? As mentioned in equation 7, the timeseries input should be from the Gaussian distribution? Is the length of time series an important hyper-parameter? Do the final points at time T (when doing classification) converge to two different stable points/attractors? Since reservoir networks always present a strong short-term memory, it would be better to discuss the connections between memory characteristic and the studied temporal kernel. Will it easily forget the information at the beginning of time series?

Correctness: Based on the a perturbative expansion the first and second order Green’s function of the system, the closed forms of the derivatives of the input projection and readout vector are derived. This process is correct and the soft margin values during optimization are also discussed in figures 2-3.

Clarity: This paper is well written.

Relation to Prior Work: Related work for reservoir networks and temporal kernel can be introduced more.

Reproducibility: Yes

Additional Feedback: Thanks to authors for the replies. I stick with my rating.


Review 4

Summary and Contributions: The paper investigates use of randomly connected RNNs for time series classification tasks, building on existing work in reservoir computing. In contrast to most reservoir computing approaches, the proposed method investigates the input weights that are traditionally drawn randomly.

Strengths: The idea to optimise input weights of a random reservoir for a binary classification task is novel. The work is also relevant to the NeurIPS community, and addresses the problem of time series classification using an efficient approach for RNN training.

Weaknesses: Weaknesses are that it is difficult to see how well we can expect the approach to work from either theoretical considerations or from a proper empirical evaluation. In the reservoir computing / ESN literature are examples of time series classification that would be useful for a comparison, for example using the Japanese Vowel Dataset is a common benchmark (eg in https://doi.org/10.1016/j.neunet.2007.04.016). It would also be good to compare against other time series classification approaches in terms of performance and compuatational complexity. The work at present does not appear to cite recent / major contributions in the area of time series classification; some relevant ones that contain further references are, for example: Wang, Z., Yan, W., & Oates, T. (2017, May). Time series classification from scratch with deep neural networks: A strong baseline. Bagnall, A., Lines, J., Bostrom, A., Large, J., & Keogh, E. (2017). The great time series classification bake off: a review and experimental evaluation of recent algorithmic advances

Correctness: The claims that the method does indeed improve time series classification would need to be backed up with more evidence.

Clarity: The paper could be improved by better separating aims, idea, derivation and realisation of the proposed idea. In the current state, it is hard to tell how the overall training procedure hsa to look like, and I would suggest to include the algorithm in the paper.

Relation to Prior Work: It becomes clear what the work is aiming to do differently. I does not become sufficiently clear how this is achieved, and neither how well the method work.

Reproducibility: No

Additional Feedback: The authors' responses clarified issues that I have previously hadn't seen, and have adjusted my overall score.

[Author Response · NeurIPS 2020]

**Question on role of the presented application:** We would like to clarify the aim of our work, as it explains the role of the chosen example application. The main motivation is to provide conceptual insight. Analytically unrolling recurrent dynamics into a (functional) Taylor series, where coefficients are given by Green's functions, is a versatile approach that may be used as a general purpose scheme. This expansion reveals how the non-linear interactions pick up higher order correlations in the input statistics, quantifying how non-linear networks provide a richer feature space than linear ones. The approach therefore elucidates which statistical features of the input data can be used by the network, thus opening a door to link and compare reservoir computing to feature-based approaches of classification.
The existence of an optimal input projection and readout vector furthermore allows one to first define and second study the performance of the recurrent reservoir itself. Thus, common methods for optimizing recurrent connectivity, including BPTT (suggested by reviewer 3), can be combined with our algorithm to study and improve the kernel properties of a reservoir network.
To illustrate the new concepts we chose one toy example that provides analytical insight and one real-world dataset to demonstrate practical applicability. We emphasize this view by the new title 'Unfolding recurrence by Green's functions for optimized reservoir computing' and agree with the reviewers that future work is required to systematically assess the performance on a broad set of problems; we are very grateful to reviewer 4 for proposing the Japanese vowel data set and the systematic approach described in Bagnall et al. 2017, that we plan to follow.

**Question on input data:** The input data does not need to fulfill any assumptions on stability, and in particular, as demonstrated with the ECG dataset, non-Gaussian stimuli are admissible. Section A.5 and A.6 in the appendix (Supplementary material) show how the soft margin is computed from the empirical moments of the stimuli, implemented in the provided code (to be published as a zenodo repository). The length of the input data is relevant for the computation of the Green's functions, which have to be computed only once before optimization and scale linearly ($G^{(1)}$) and quadratically ($G^{(2)}$) with the stimulus length. All summations over time indices can be performed prior to the actual optimization, which thus becomes independent of the length of the data. Centering of data is possible without knowledge of the labels and can also be avoided by incorporating a threshold in eq. (4) and following derivations. The scaling of inputs is arbitrary, as it can be absorbed by rescaling $\alpha$. Both were performed here only for conceptual clarity, allowing identical network parameters for both datasets and to simplify the presentation of the mathematical details. The supplementary material furthermore contains a description of the full algorithm. The required level of detail to ensure reproducibility unfortunately prohibited the inclusion in the main text. We will point more clearly to the appendix and also include pseudocode in a revision.

**Question on stability of dynamics, chaos, memory life time:** While the Green's function of the linear system remains well-defined also in the linearly unstable regime (spectral radius of $W$ exceeding unity; chaotic dynamics), the perturbative solution of the non-linear system built thereof (eqs. (9)-(12)) suffers from exponentially growing modes. We are currently working on a multi-timescale approach that propagates only for short temporal intervals and then recomputes the Green's functions; it appears to work reliably in the chaotic regime. Generally, the initial network state should not be chosen too large as the approximate solution of the Green's function requires operation in the near-linear regime; otherwise, it is irrelevant. The final network states do not converge to fixed points of the dynamics: If the network was evolved beyond $t > T$, the states would continue changing. In this regard, the presented work is drastically different from the view of computation by fixed-points and slow points (see e.g. review by Susillo et al. 2014), to be discussed in the revision.
The memory characteristic reviewer 3 was concerned with can be seen from the slowest decaying mode with time constant $\tau/(1 - \max \text{Re}(\lambda_\alpha))$, where $\lambda_\alpha$ are the eigenvalues of $W$. For spectral radii close to unity, the time scale thus becomes very long (diverging); otherwise the network forgets exponentially in time. The statements by reviewer 2 on this topic remain true for optimized input projection. We pointed out the exponential forgetting, because in the presented applications the decision is made at the final time point. This means that there is room for improving performance in particular in the ECG dataset, either by using a reservoir closer to instability or by a different readout mechanism (e.g. integrated over time).

**Question on non-linearity:** For conceptual clarity we assumed a weak ($\propto \alpha \ll 1$) nonlinearity to enable a vanilla perturbative expansion. For stimuli with low linear separability, even small nonlinearities significantly increase classification performance (see artificial stimulus, Fig. 3b). This is, however, not the case for the chosen real-world application as the latter is already easy to separate linearly (giving rise to the same accuracies in Table 1, discussed in lines 235-36 and 241-44). Future work is left to assess the performance of weakly non-linear reservoirs on other real-world data. In addition, stronger, arbitrary non-linearities can be handled by the multi-timescale approach (see above), or by established methods: hard non-linearities (Heaviside; binary / spiking networks) allow the use of the Gram-Charlier expansion (Dahmen et al. 2016 PRX; Farkhooi et al. 2017 PRL), exploiting that intrinsically-generated network noise smooths out hard non-linearities, to be discussed in the revision.

**Further comments:** We will take care of the feedback by reviewers 1, 3 and 4 to improve the embedding into current literature and gratefully acknowledge also all minor points that are very helpful to revise the manuscript.

[Meta-Review · NeurIPS 2020]

The paper is interesting and provides a new line of thinking about reservoir modeling through soft margin with an interpretation in terms of cumulants and second order Green function. The work serves as a proof-of-concept that perturbation analysis of the interaction is useful in practical computational problems. Reviewers recommend acceptance, but we strongly urge the authors to address the issues raised by the reviewers and add discussions to shed more light on intricate details and provide insights. The authors' response addresses some of these issues already, and this should be integrated into the final manuscript.